# Changes in Rheological Properties of Mortars with Steel Slags and Steel Fibers by Magnetic Field

**DOI:** 10.3390/ma14144005

**Published:** 2021-07-17

**Authors:** Dukman Kang, Doyoung Moon, Wooseok Kim

**Affiliations:** 1Dong-A ENG, 592, Cheonho-daero, Gwangjin-gu, Seoul 04986, Korea; ace-kang@naver.com; 2Department of Civil Engineering, Kyungsung University, 309 Sooyoung-ro, Nam-gu, Busan 48434, Korea; 3Department of Civil Engineering, Chungnam National University, 99, Daehak-ro, Yuseong-gu, Daejeon 34134, Korea

**Keywords:** magnetic field, steel fibers, rheology

## Abstract

The effect of a magnetic field on the rheology of mortars with steel slags and fibers was evaluated in this study. The rheology of the mortar measured with and without a magnetic field was compared. The effect of steel fiber addition to normal and steel slag mortars, mix ratio and size of steel fibers, and magnetic field formation position on rheology were evaluated. Steel fiber addition increased the yield stress and viscosity of the normal and steel slag mortars. The increased rheology was almost restored because of the magnetic field applied to the normal mortars. However, the increased rheology of the steel slag mortars with steel fibers was restored only upon the application of the magnetic field, whose position was continuously changed by a power relay. It is deduced that the alignment of the steel fibers by the magnetic field contributes to the rheology reduction of the mortars. However, in the case of steel slag mortar, experimental results demonstrated that steel slag, which is a ferromagnetic material, receives constant force by the magnetic field, which increases the rheology. This is evidenced by the decrease in the rheology of steel slag mortars under a continuously changing magnetic field formation position by power relay.

## 1. Introduction

Steel fiber is used to reinforce the insufficient toughness of concrete. Steel fibers reinforced in concrete are well-known to significantly contribute to preventing crack widening by bridging cracks [1,2,3]. Additionally, steel fiber is frequently used for tunnel segments, mainly in Europe, and to reduce the damage caused by the jack thrust of tunnel boring machines [4].

The orientation of steel fibers in steel fiber-reinforced concrete plays a dominant role in the successful application of steel fiber-reinforced concrete described above. The flexural crack resistance of steel fiber-reinforced concrete was significantly affected by the fiber orientation coefficient. According to the experimental results obtained by other researchers, the more steel fibers oriented in the direction perpendicular to the crack, the higher the crack resistance [5,6]. The orientation of the steel fiber has a significant influence on the toughness, which increases as the number of steel fibers perpendicular to the fracture surface increases. The post-peak fracture energy is defined as a function of the steel fiber orientation coefficient [7]. Even if the same number of steel fibers is used, a better reinforcing effect can be obtained when the number of steel fibers aligned perpendicular to the crack is significant [8]. Therefore, considerable research has been conducted on the evaluation of the orientation and distribution of steel fibers in concrete, the correlation between the orientation of the steel fibers and the engineering characteristics of fresh concrete, and a mix design to enhance the workability, consistency, and stability of steel fiber-reinforced concrete [9,10,11].

Since 2015, researchers have attempted to use a magnetic field to align the direction of steel fibers mixed in concrete and mortar [12,13,14,15,16,17,18,19,20,21,22]. This method corresponds to the most active research attempt to artificially align the direction of steel fibers in concrete. In these studies, fresh-state steel fiber-reinforced concrete was exposed to a bobbin wrapped with a large number of coils so that the direction of the steel fibers was aligned in a direction perpendicular to the coil wrapping direction. The aligned steel fibers in concrete were analyzed using image analyses such as a CT scan, inductive method, and visual inspection of steel fibers at the fractured surface. Improvements owing to the aligned steel fibers were shown in the increases in the flexural strength, splitting strength, and fracture energy.

Research results have suggested that a magnetic field is effective not only for aligning the steel fibers in concrete but also in driving and vibrating concrete and mortar in which steel slags or iron sand are partially replaced. Chen et al. reported that magnetically driven concrete, in which half the aggregate was replaced with steel slags, indicated that concrete compacted with a magnetic field had better strength than concrete compacted with conventional vibration [13]. In their experiment, compaction using a magnetic field was used to manually control the magnetic field at 3 s intervals and compare it with conventional vibration. The compressive, splitting, and flexural strengths were higher, and the porosity from the mercury intrusion porosimetry test was lower upon magnetic field compaction than vibration compaction.

Additionally, Xue et al. evaluated the orientation of steel fibers and the flexural strength of steel fiber-reinforced mortar and concrete in which aggregates were partially replaced with steel slag and iron sand [21]. The direction of the steel fibers in the concrete and mortar compacted with a magnetic field were well aligned, and the flexural strength was higher than that of those compacted with a shaking table conventionally used in their experimental results.

However, in these papers, only improvements in efficiency and strength when using a magnetic field as a compaction method were presented, but the evaluation of fresh-state mortar and concrete and explanation for the improved results were not presented. An analytical model capable of evaluating and analyzing the effect of a magnetic field on the alignment of steel fibers mixed in a fresh-state mortar was presented by Villar et al. [23,24,25]. Through an experimental program using one steel fiber attached to the vane-type rheometer, the correlation between the rotational force acting on the steel fibers owing to the magnetic field and the resistance owing to the mortar suspension was analyzed, and an analytical model was presented for evaluation and prediction.

The rheological properties of fresh concrete are used as indicators of consistency, workability, and flowability, and are used to predict the stability, pumpability, shootability, and pressure of the formwork. According to research on the effect of steel fibers on the rheology of steel fiber-reinforced concrete, the addition of steel fibers increased the yield stress and plastic viscosity as the amount of steel fiber increased, the aspect ratio of the fiber increased, and the diameter of the fiber increased [26,27]. However, in experimental results of the effect of the rheology of steel fiber-reinforced self-compacting concrete on the direction and distribution of steel fibers, it was found that the lower the viscosity, the more steel fiber segregation occurred [28].

In this study, the effect of the electromagnetic field on the rheology of steel fiber-reinforced mortar was examined. The rheology of normal and steel slag mortars and those with steel fibers were measured using a vane-rotation-type rheometer. The electromagnetic flux density, magnetic field with and without power relay, amount of steel fiber mixed, and steel fiber size were tested as variables. The experimental results were analyzed in terms of the rate of change of the rheology with and without the application of a magnetic field.

## 2. Experimental Program

The experimental program evaluated the effect of the magnetic field on the rheological properties of normal and steel slag mortars with steel fibers. The program was conducted in two phases. In the Phase I test, the effect of the magnetic field on the yield stress and viscosity change of the normal and steel slag mortars was evaluated with and without a magnetic field. In the Phase II test, an experiment was conducted to determine the difference in the effect of the magnetic induction method on the rheology of steel slag mortars.

Here, the magnetic induction method was tested in two ways. The first maintained a constant magnetic field over the entire height of the sample container during the measurement of rheology, and the second divided the coil into the top, middle, and bottom of the container and continuously changed the position of the magnetic field formation during the measurement of rheology using a power relay. Brookfield DV3T, which is a vane-rotation-type rheometer, was used to evaluate the rheological parameters (yield stress and plastic viscosity) of the normal and steel slag mortars with and without steel fibers.

### 2.1. Mix Proportions

In this experiment, the rheology of the four formulations was measured. Table 1 lists the mix proportions of the mortars. The water–cement ratio for all mortars was 0.42. In the table, NM is a mixture of normal mortar, SSM is a mixture of 100% steel slag instead of sand, and NM_SF is a mixture in which 0.25% vol. or 0.50% vol. of steel fibers are added to the NM. SSM_SF is a mixture in which 0.25% vol. of steel fibers was added to the SSM.

Compressive strength tests were performed at 7 days and 28 days after casting. The strength test results are presented in Table 1. It can be seen that the strength of the mortar reinforced with steel fibers is approximately 5 MPa greater.

### 2.2. Materials

CEM I Portland cement was used in this study. Siliceous sandwith a maximum particle size of 2 mm and steel slag were used. The material properties of sand and steel slag are listed in Table 2. The surface dry density and moisture absorption were tested according to KS F 2504 and fineness modulus according to KS F 2502. The other properties were provided by the manufacturer. The particle size distributions of the sand and steel slag are listed in Table 3. The chemical composition of steel slag varies depending on the composition of the raw material in the iron production process, but it has approximately 40% calcium oxide (CaO), 35% silica, 13% alumina, and 8% magnesia, according to the manufacturer’s report. Figure 1a shows the steel slag contained in a glass beaker and the steel slag attached to the magnet in response to the magnet.

The ferromagnetic steel fibers used in this experiment are micro steel fibers and twisted steel fibers, referred to as SF1 and SF2, respectively. These are presented in Table 4. Figure 1b shows the steel fibers used in this study. The dimensions and tensile strengths of the steel fibers used are listed in Table 4.

### 2.3. Magnetic Field

Cylindrical plastic containers used as bobbins were manufactured by 3D printing. The containers have a height of 150 mm and inner and outer diameters of 75 mm and 80 mm, respectively, and are suitable for the rheology measurement of mortars. A magnetic field was generated by winding a 0.8 mm diameter coil on the outer surface of the plastic container. Figure 2a,b show the appearance of the container before winding the coil and the container with insulating tape attached to the surface of the wound coil, respectively. For this experiment, three plastic containers with different winding coils were used, as listed in Table 5. The number of coil windings on containers 1 and 3 is the same as 2500 turns, and the number of coil windings on container 2 is 1500.

However, the number of coils wound was different. Only one coil was wound on container 1, but three coils were wound on container 3. It was wound 750 turns on the top and bottom of the container, and 1000 turns on the middle. The coils wound on the top, middle, and bottom have overlapping parts, as shown in Figure 3. The electrical resistivity of the coils wound on each container was measured using a digital multimeter and is listed in Table 4. The resistance of the coil wound on container 1 is approximately 10 Ω greater than the resistance of the coil wound on container 2. Therefore, a greater electromagnetic flux density can be induced in container 2 than in container 1 at the same voltage.

In the experiment using containers 1 and 2, in which one coil was wound over the entire height of the container, a constant electromagnetic force was generated over the entire height of the containers. For the experiment using container 3, a power relator was used, as shown in Figure 4. It was programmed to relay power to three coils wound on container 3 at 1 s intervals. Therefore, the position of generating the magnetic force induced in container 3 continuously changes to the upper, middle, and lower positions every second. It is possible to compare the difference between the constant magnetic force formation and the change in the magnetic force formation position using the relator.

The coils are connected to an AC power supply. This power supply has the capacity to apply 10 to 240 V, but 50 and 90 V were applied in the experiment. The voltage applied to the coil in this experiment was determined from the analysis of the equilibrium equation which is presented in [15,17], between the magnetic force to rotate and the resistive force to resist the rotation of the steel fiber in the cement matrix. Among the two, 50 V was adopted in the range where the alignment of the steel fibers can bring a definite change in the rheology of the mortar. 50 V induces a magnetic force sufficient to vertically align almost all the steel fibers in the mortar. Also, in the case of 90 V, the highest applicable voltage is adopted within the range that allows safe experiments. This is because voltages higher than this will cause the coil temperature to rise excessively, causing the coil to melt or catch fire.

The amount of current passing through this coil was measured using a multimeter and is presented in Table 5. The magnetic flux density was measured using a Gauss meter that can measure up to 1 T at the top center of the container, as shown in Figure 2c. The results are presented in Table 5. The measured magnetic flux density is 0.0025–0.02 T. It is considered that this level of density is sufficient to rotate the steel fibers in the mortar by 90° based on the experimental results presented in [24]. The power consumption per unit volume of mortar for this magnetic method is presented in Table 5.

### 2.4. Test Procedure and Rheological Properties

The experiment was conducted in two phases. In the Phase I test, normal and steel slag mortars with and without steel fibers were tested in container 1. The effect of the magnetic field on the changes in the yield stress and plastic viscosity of the normal and steel slag mortars with and without steel fibers was primarily evaluated. The mix ratio and size of the steel fibers were varied. Moreover, rheology changes owing to the magnetic field in different suspensions, namely, normal mortar and steel slag mortar, were compared with each other. In the Phase II test, steel slag mortar with and without steel fibers was tested in containers 2 and 3, respectively. A greater flux density was examined through the test using container 2, and the effects of constant magnetic force formation and the change in the magnetic force formation position by using the relator was evaluated through the test using container 3, as summarized in Table 6.

Rheology was measured in all experiments using the following procedure: shortly after casting the mortars, the samples were immediately transferred to the inside of the containers. After adjusting the height to sufficiently submerge the vane of the rheometer, the rotating speed of the vane was controlled over time, as shown in Figure 5.

This pre-shear cycle process ensures that the mortar sample has the same shear history for all mortars tested. In the data logging cycle, one data point was acquired every 5 s, and the dynamic yield stress and plastic viscosity were determined from these results. The Bingham model was adopted to fit the shear stress-shear rate data of the down curves. The Bingham model can be expressed as (1), where $\tau_{0}$ is the dynamic yield stress, and μ is the plastic viscosity. τ and γ are the shear stress and strain rate, respectively, in Equation (1):(1)$$\tau =\tau_{0}+\mu \gamma$$

Rheological properties were measured twice for the same mortar samples. The first was measured without passing electricity through the coil, and the second was measured while passing electricity through the coil. To distinguish between them, the first measurement result is specified in Table 7 as “w/o,” and the second measurement result is specified in Table 7 as “w.” “w/o” is an abbreviation of “without,” meaning that it was measured without a magnetic field. “w” stands for “with,” meaning that it was measured after the formation of a magnetic field. Immediately after the first measurement, a second measurement was taken, but the sample was stirred 10 times with a rod before the second measurement to remove the shear yield surface that might have occurred with the first measurement.

The test mortar groups and IDs are summarized in Table 6. The name of the test group of mortars was named in the order of the type of mortar (NM or SSM), type of steel fiber and its mixing amount, and applied voltage. For example, NM_SF1 (0.25)_50 is a normal mortar suspension. The SF1 steel fiber was mixed at a volume ratio of 0.25%, and the applied voltage was 50 V. Figure 6 shows the test setup for the rheology measurement of the mortar under a magnetic field.

## 3. Results

The yield stress and viscosity values obtained from the experimental program are listed in Table 7. The rate of change in yield stress and viscosity owing to the magnetic field was calculated using Equation (2):(2)$$\mathrm{Change}\;\mathrm{rate}:\;\frac{\omega -\omega /ο}{\omega /ο}\times 100$$

The mean and standard deviation of yield stress of NM_SF1 (0.25), NM_SF1 (0.50), and NM_SF2 (0.25) measured without a magnetic field (“w/o” in Table 7) for the Phase I test were 20.69 ± 0.23 Pa, 26.08 ± 0.45 Pa, and 21.95 ± 0.16 Pa, respectively. However, SSM and SSM_SF measured without a magnetic field (“w/o” in Table 7) for the Phase I and Phase II tests were 14.85 ± 0.16 Pa, and 18.89 ± 0.14 Pa, respectively. In addition, the average and standard deviation of the viscosity of NM_SF1 (0.25), NM_SF1 (0.50), and NM_SF2 (0.25) were 1.33 ± 0.01 Pa·s, 2.14 Pa·s ± 0.04 Pa·s, and 1.88 ± 0.01 Pa·s, respectively. The average and standard deviation of the viscosity of SSM and SSM_SF were 1.60 ± 0.06 Pa·s and 1.92 Pa·s ± 0.10 Pa·s, respectively. The standard deviation of all experimental results was only 5% at the maximum, indicating that the experiment was properly performed without any significant differences in the experimental environment.

Figure 7 compares the average yield stress and viscosity measured without a magnetic field for all formulations tested. It was found that the mixing of steel fibers significantly increased the yield stress and viscosity. The same trend was observed for NM and SSM. In the case of NM, it was found that the higher the fiber mixing amount and the larger the steel fiber, the greater this tendency. Compared to NM, the yield stress of SSM was 6% smaller, but the viscosity was 29% larger. The results reflect the characteristics of the fine aggregates listed in Table 2.

The increase in yield stress and viscosity owing to steel fiber mixing was also confirmed in Lasker et al.’s experiment [26]. In their rheology measurement results for steel fiber-reinforced concrete, the yield stress increased from 2 to 4 times and the viscosity increased from 1.1 to 2.1 depending on the mixing ratio of steel fibers. According to [29], the rotation and orientation of the slender fibers during the flow is affected and/or countered by neighboring particles and fibers. Rheology is consequently influenced by mechanical contacts between fibers and hydrodynamic effects between fibers and neighboring particles.

### 3.1. Results of Phase I Test

Figure 8, Figure 9 and Figure 10 show the experimental results for NM and NM with steel fibers. NM showed little change in the yield stress and viscosity with and without a magnetic field. Since there is no ferromagnetic material in the NM formulation, this change could originate from the difference in experimental conditions. Figure 8 clearly shows the decrease in yield stress and viscosity owing to the effect of the magnetic field at 50 and 90 V. However, it seems that the decrease in the rate of change in stress and viscosity is not proportional to the voltage. The rate of change of yield stress is not significantly different between 50 and 90 V. The rate of change in viscosity is sometimes larger at 50 V. Therefore, the magnetic field intensity at 50 V is considered to have a sufficient effect on the reduction in rheology. This is because most of the increased rheology owing to the addition of steel fibers was found to be reduced to the rheology of NM at this level of magnetic field intensity.

Figure 9 shows the change rate of rheology according to the mixing ratio of the steel fibers. When the mixing amount of steel fiber is 0.5% vol., the decrease in yield stress and viscosity is almost twice and triple the decrease when the mixing amount is 0.25% vol.

Figure 10 shows the influence of the type of steel fiber. As described above, SF1 is a micro steel fiber, and SF2 is a commonly used macro steel fiber. It was found that the same amount of steel fibers was mixed, and the aspect ratio of the steel fibers was almost the same, but the decrease in rheology owing to the magnetic field was different. At 50 and 90 V, the yield stress of SF1 decreased by 16.76% and 21.68%, while that of SF2 decreased by 30.93% and 34.07%, respectively. The decrease in viscosity also showed a similar trend. As shown in Figure 7, the greater the amount of steel fiber mixed and the greater the size of the steel fiber, the greater the rheology. It should be noted that the reduction in rheology owing to the magnetic field is also very similar to this result.

Figure 11 and Figure 12 show the test results for the steel slag mortar. Figure 11 compares the results of the SSM according to the voltage. As shown in Figure 11, even if the magnetic force increases owing to an increase in voltage, the viscosity does not significantly decrease, and the yield stress increases. Figure 12 shows the SSM results compared with the results of the NM series. Owing to the magnetic field, the rheology decreases in the NM_SF series while it increases in the SSM and SSM_SF series at both 50 and 90 V. When comparing SSM and SSM_SF, the increase in SSM_SF is insignificant compared to the SSM. Because steel slag is a ferromagnetic material, it was expected to contribute to the reduction of rheology by reacting to magnetic fields such as steel fibers, but unexpected results were obtained from steel slag mortar tests. This insignificant effect of steel fibers in steel slag mortar compared to in normal mortar was further investigated in the Phase II test, in containers with a greater magnetic flux density and with a relator, respectively.

### 3.2. Results of Phase II Test

Figure 13 shows the yield stress results of SSM and SSM_SF1 (0.25) from the Phase II test. The yield stress decreases in container 1 but increases in container 2.

This trend is the same at 50 and 90 V. The yield stress decreases in container 1 but increases in container 2. In container 3, a large reduction in the yield stress is observed. This trend is the same at 50 and 90 V. In addition, in the results of container 1, there is a slight difference between SSM and SSM_SF, but there is a significant difference between containers 2 and 3. In Figure 14, the viscosity of the SSM increases in containers 1 and 2 but decreases in container 3.

The viscosity of SSM_SF1 (0.25) was not significantly reduced in containers 1 and 2 but decreased significantly in container 3. This result demonstrates that the rheology of SSM and SSM_SF is significantly affected by the application method of the magnetic field. In particular, in container 3, because the application position of the magnetic field is relayed over time, the magnetic force that pulls the steel slag and the steel fiber acts according to the position relayed, resulting in an increase in the fluidity of the SSM and a consequent decrease in the yield stress and viscosity.

## 4. Discussion

The magnetic field applied to container 1 significantly reduced the rheology of the normal mortar with steel fibers so that the rheology of the normal mortar with the steel fibers became almost the same as that of the mortar without steel fibers. However, this effect was not observed in the results for the steel slag mortar and steel slag mortar with steel fibers. The magnetic field is considered to contribute to increasing the rheology of steel slag mortar and steel slag mortar with steel fibers. However, a significant reduction in the rheology of steel slag mortar and steel slag mortar with steel fibers occurred when container 3 was used.

The decrease in the rheology of the normal mortar with steel fibers is related to the vertically aligned steel fibers owing to the magnetic field (see Figure 15). In high-viscosity blending, the interaction between steel fibers is regarded as one of the main factors that increase the viscosity [29]. However, in this experiment, the interaction between the steel fibers decreased as the steel fibers were vertically aligned by the magnetic field, which is believed to have had a significant effect on the reduction in rheology.

Figure 16 shows the causes the incrementing rheology of steel slag mortar and steel slag mortar with steel fibers obtained from the tests using container 2. The figure shows container 2 before and after 90 V passes through; the container has steel fibers and steel slags. As shown in the figure, the steel slag and steel fibers are receiving vertical force under a magnetic field and are arranged vertically as a column to maintain their shape. This leads to an increase in the yield stress and viscosity. However, as in the test using container 3, if the magnetic field formation position is periodically relayed, then it is believed that the steel slag is vertically disturbed in a short time, resulting in a decrease in the rheology.

## 5. Conclusions

In this study, changes in the rheology of normal and steel slag mortar with steel fibers were tested under a magnetic field using a vane-rotation-type rheometer. The conclusions obtained from the experiment are as follows:

(1) The increase in rheology caused by the addition of steel fibers in normal mortar was almost restored to the rheology of normal mortar without steel fibers by adjusting the direction of steel fibers through the magnetic field. The adjusting of steel fibers in normal mortar owing to magnetic field reduced the interaction between the steel fibers mixed in the mortar.

(2) The experimental results of the steel slag mortar under a magnetic field showed a significant difference from those of normal mortar. Moreover, the results changed significantly depending on the power relation and the formation position of the magnetic force. The rheology of steel slag mortar and steel slag mortar with steel fibers increased under a constant magnetic field formation and decreased under continuous change in the formation of the magnetic field using a power relator. Steel slag forms columns by a magnetic field in the steel slag mortar, and the force required to maintain the columns was sustained by the magnetic field during the test. It is believed that the force could be the reason for the increased rheology of the steel slag mortar and steel slag mortar with steel fibers. This result suggests that it is more feasible to continuously change the position of the magnetic force rather than continuously act on the same position to reduce the rheology of steel slag mortar and steel slag mortar with steel fibers.

Generally, rheology is used as an indicator to evaluate workability, flowability, stability, pumpability, and shootability. Therefore, an additional experimental program is recommended to understand whether the change in the rheology of mortars containing steel slag and fibers caused by magnetic fields contributes to the change in these abilities.

## Figures and Tables

**Figure 1 materials-14-04005-f001:**
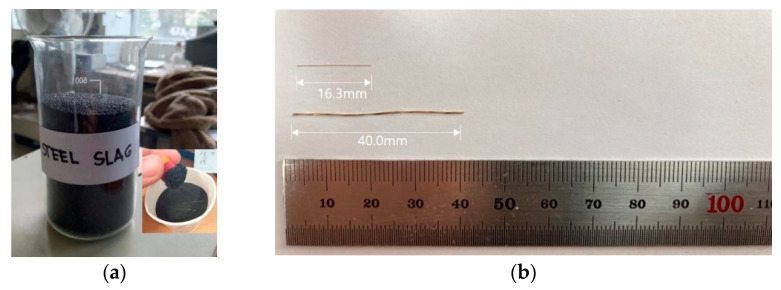
Slags and steel fibers used for experiment: (**a**) steel slags, (**b**) steel fibers.

**Figure 2 materials-14-04005-f002:**
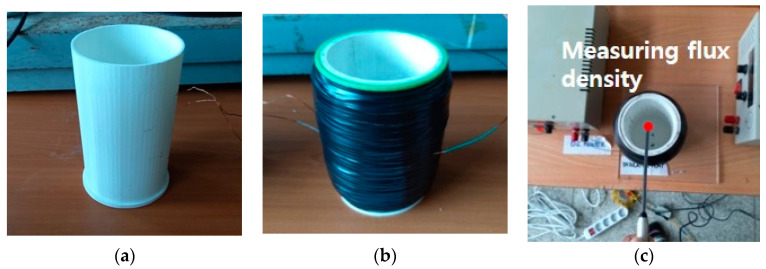
Plastic container for magnetic field: (**a**) before winding coils, (**b**) after winding coils, (**c**) measuring location of flux density.

**Figure 3 materials-14-04005-f003:**
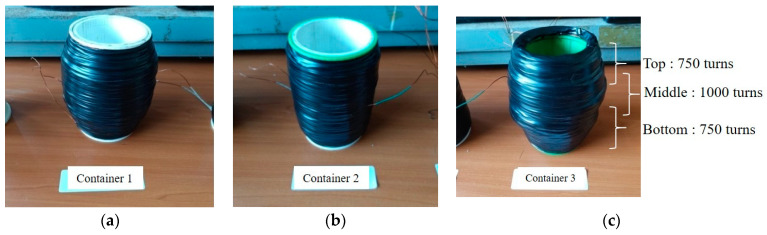
Plastic containers used; (**a**) container 1, (**b**) container 2, (**c**) container 3.

**Figure 4 materials-14-04005-f004:**
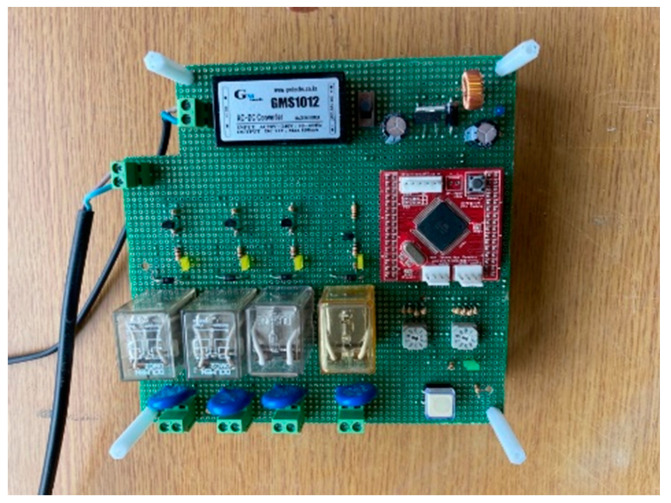
Power relator used for container 3.

**Figure 5 materials-14-04005-f005:**
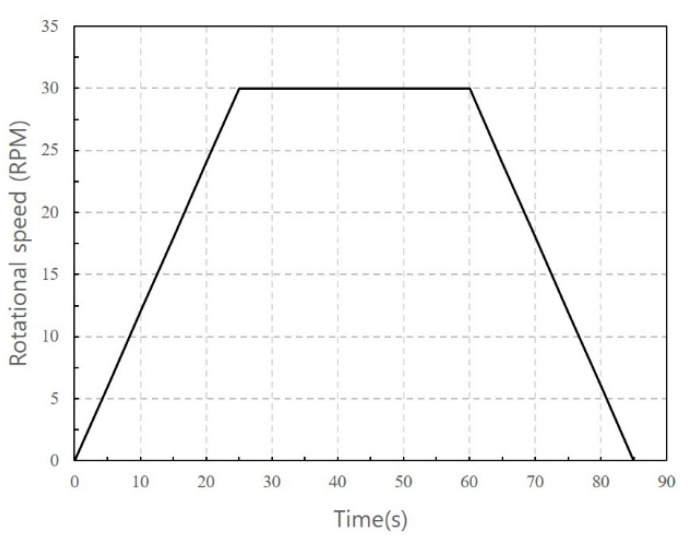
Paddle control program for rheology test.

**Figure 6 materials-14-04005-f006:**
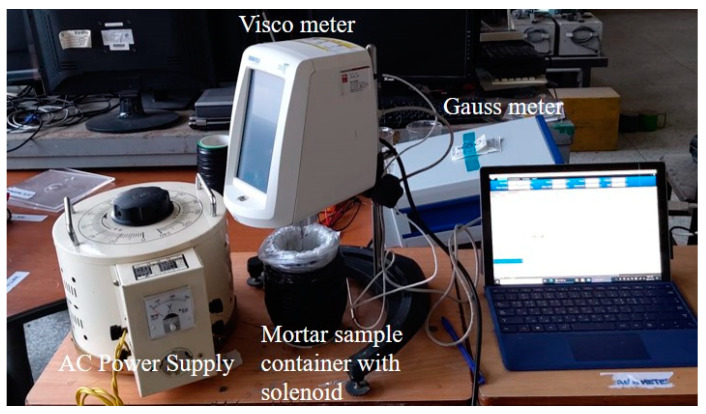
Test setup.

**Figure 7 materials-14-04005-f007:**
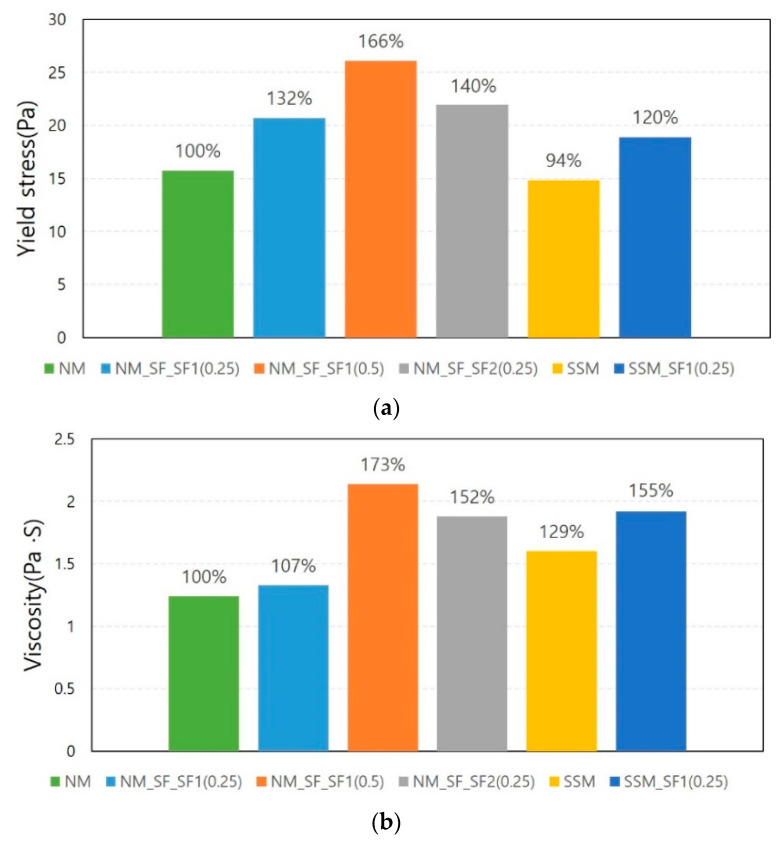
Average yield stress and viscosity measured without magnetic field: (**a**) yield stress, (**b**) viscosity.

**Figure 8 materials-14-04005-f008:**
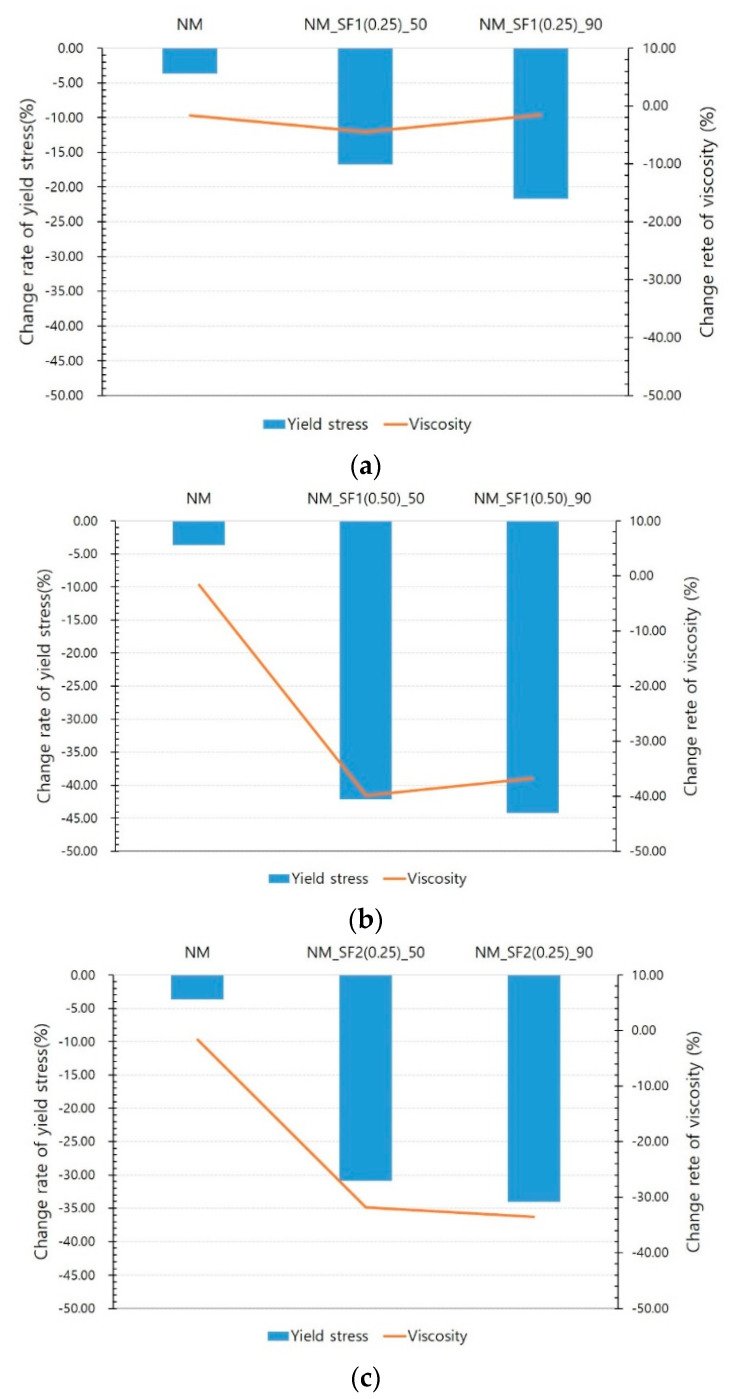
Effect of magnetic flux density: (**a**) NM and NM_SF1 (0.25), (**b**) NM and NM_SF1 (0.50), (**c**) NM and NM_SF2 (0.25).

**Figure 9 materials-14-04005-f009:**
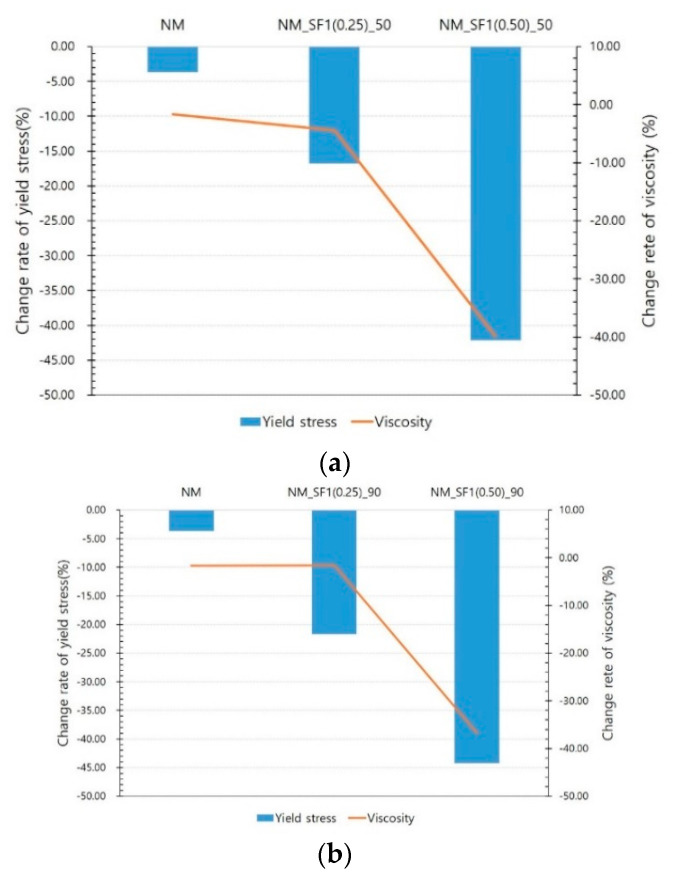
Effect of steel fiber mix ratio: (**a**) at 50 V, (**b**) at 90 V.

**Figure 10 materials-14-04005-f010:**
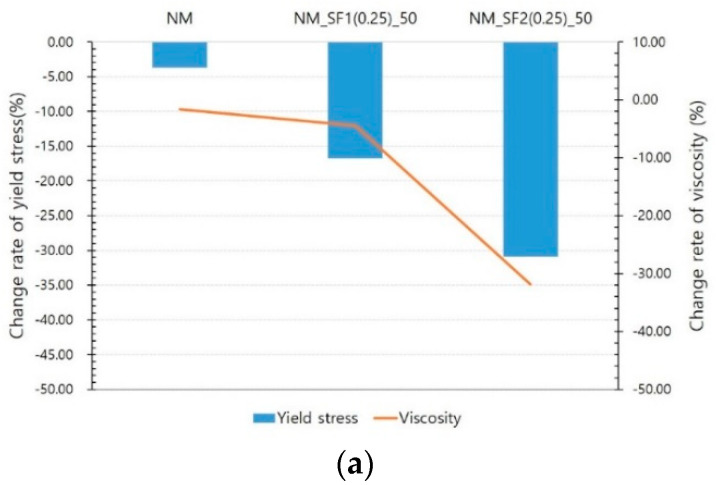
Effect of steel fiber type: (**a**) at 50 V, (**b**) at 90 V.

**Figure 11 materials-14-04005-f011:**
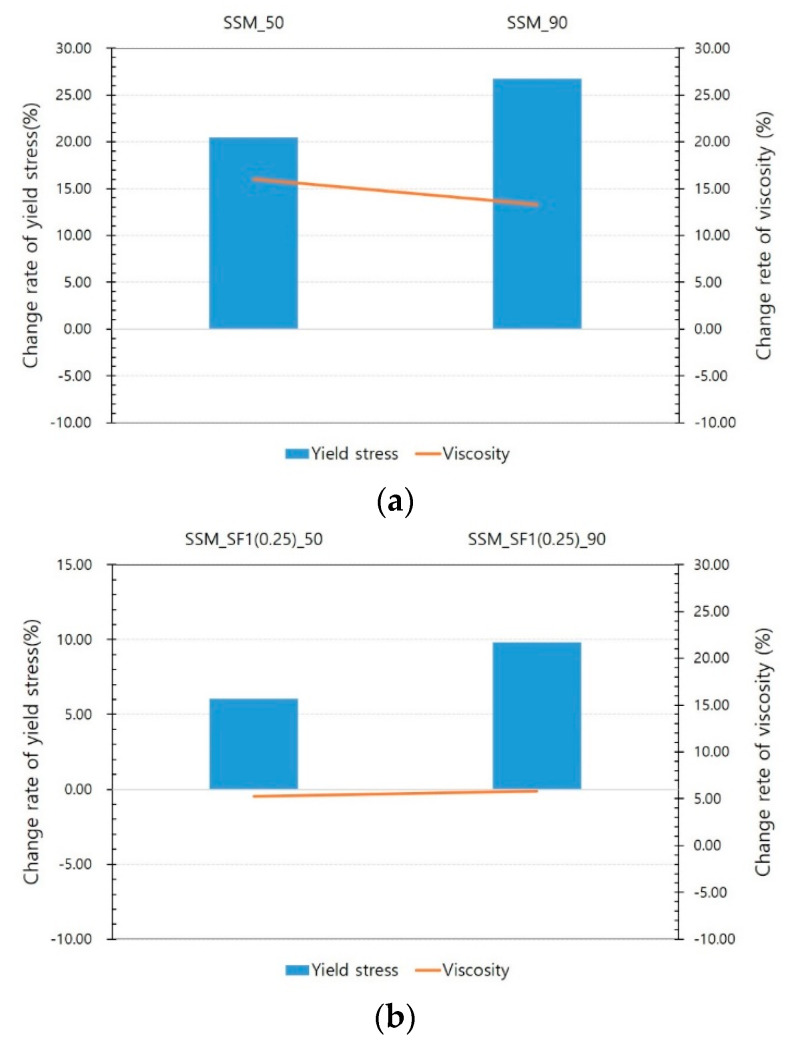
Results of SSM according to applied voltage: (**a**) SSM, (**b**) SSM_SF1(0.25).

**Figure 12 materials-14-04005-f012:**
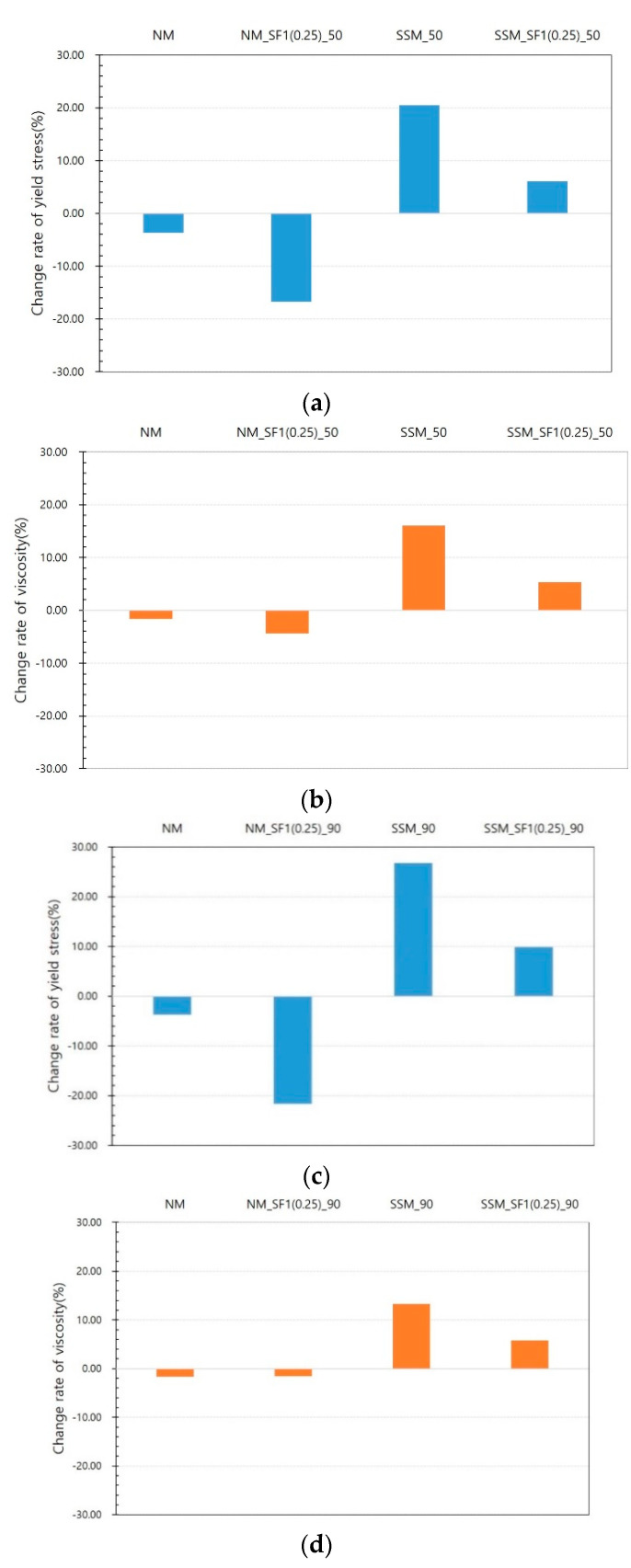
Comparison between normal and steel slag mortars: (**a**) yield stress at 50 V, (**b**) viscosity at 50 V, (**c**) yield stress at 90 V, (**d**) viscosity at 90 V.

**Figure 13 materials-14-04005-f013:**
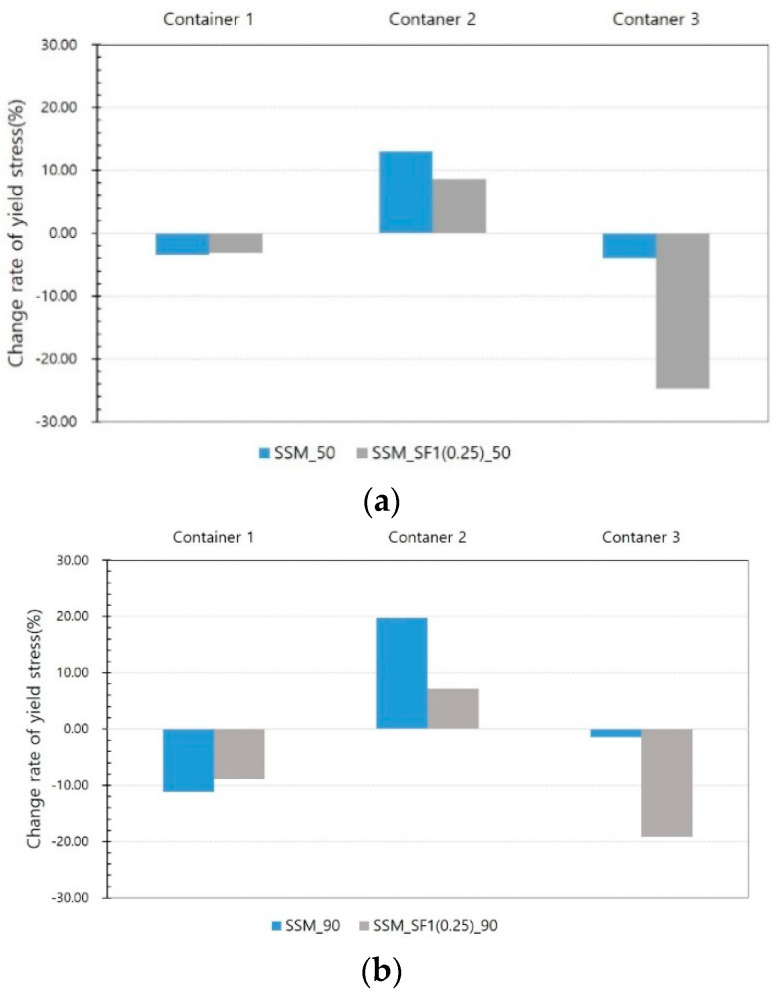
Yield stress results from Phase II test (**a**) at 50 V, (**b**) at 90 V.

**Figure 14 materials-14-04005-f014:**
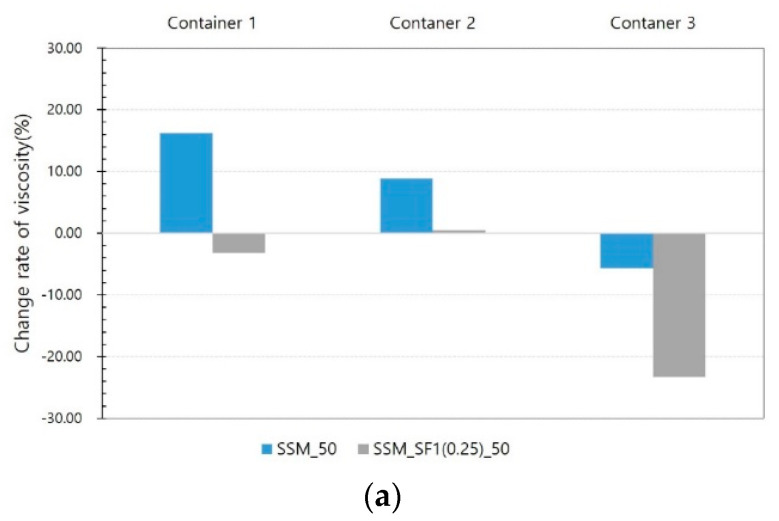
Viscosity results from Phase II test: (**a**) at 50 V, (**b**) at 90 V.

**Figure 15 materials-14-04005-f015:**
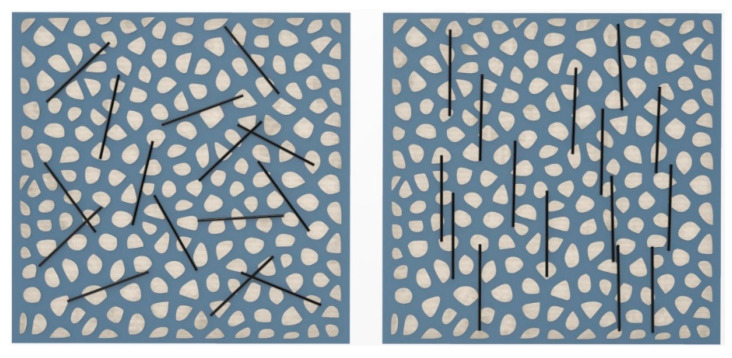
Vertically aligned steel fibers owing to magnetic field in container 2.

**Figure 16 materials-14-04005-f016:**
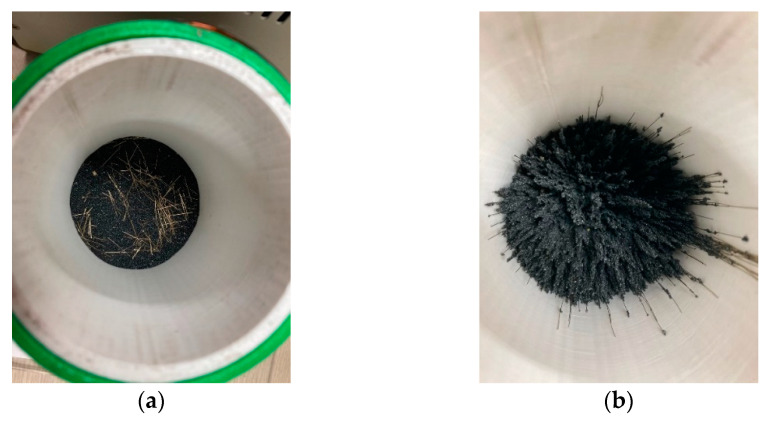
Steel slag and steel fibers under magnetic field: (**a**) before applying voltage, (**b**) after applying voltage.

**Table 1 materials-14-04005-t001:** Mix proportions and compressive strengths of mortars.

Group	Cement, kg/m^3^	Water, kg/m^3^	Sand, kg/m^3^	Steel Slag, kg/m^3^	Steel Fiber, kg/m^3^ (%Vol.)	Compressive Strength (MPa)	
						at 7 Days	at 28 Days
Normal Mortar (NM)	477	200	620	-	-	21.86 ± 3.06	29.11 ± 4.06
Steel Slag Mortar (SSM)	477	200	-	620	-	22.40 ± 4.52	29.79 ± 6.01
Normal Mortar with Steel Fibers (NM_SF)	477	200	620	-	20(0.25)	24.80 ± 3.52	34.80 ± 5.27
Steel Slag Mortar with Steel Fibers (SSM_SF)	477	200	-	620	20(0.25)	25.20 ± 5.01	35.62 ± 5.65

**Table 2 materials-14-04005-t002:** Material properties for fine aggregates used.

	Absolute Dry Density (g/cm^3^)	Surface Dry Density (g/cm^3^)	Moisture Absorption (%)	Fineness Modulus	Mass Density (kg/m^3^)
**Steel slag**	3.56	3.57	0.42	3.16	2263
**Sand**	2.57	2.58	0.60	3.98	1575

**Table 3 materials-14-04005-t003:** Particle size distribution of fine aggregates.

	Size (mm)	<0.075	0.075	0.15	0.3	0.6	1.18	2.36	4.75
**Steel** **slag**	**Retained** **(%)**	80.0	79.9	79.5	76.7	53.7	23.6	3.0	0
**Sand**		100	99.6	85.3	52.6	31.7	10.7	0.7	0

**Table 4 materials-14-04005-t004:** Physical parameters of fibers used.

ID	Shape	Diameter (mm)	Length (mm)	Aspect Ratio	Tensile Strength (MPa)
**SF1**	Straight	0.2	16.3	81.5	1500
**SF2**	Twisted	0.5	40.0	80.0	1100

**Table 5 materials-14-04005-t005:** Details of winding of coils and measured current and magnetic flux density for each container.

	Number of Turns	Number of Coil	Resistivity (Ω)		Applied AC Volt (V)	Current (A)	Power Consumption (Watt/m^3^)	Magnetic Flux Density (G)
**Container 1**	2500	1	25.2		50	0.28	21.14	25.54
	2500	1	25.2		90	0.63	85.61	54.81
**Container 2**	1500	1	15.7		50	1.07	80.77	59.12
	1500	1	15.7		90	2.08	282.63	128.18
**Container 3**	2500	3	top	7.2	50	2.33	175.89	92.46
	2500	3	top	7.2	90	1.13	153.55	231.79
	2500	3	middle	11.5	50	2.30	173.63	98.34
	2500	3	middle	11.5	90	4.90	665.82	203.85
	2500	3	bottom	7.2	50	2.55	192.50	96.90
	2500	3	bottom	7.2	90	4.56	619.62	146.08

**Table 6 materials-14-04005-t006:** Summary of test mortar groups.

	Mortar	Steel Fibers		Applied Volts (V)	Test Groups	Container
		Type	Vol. (%)			
Phase I	NM	-	-	50	NM	Container 1
		SF1	0.25	50	NM_SF1(0.25)_50	
				90	NM_SF1(0.25)_90	
		SF1	0.50	50	NM_SF1(0.25)_50	
				90	NM_SF1(0.50)_90	
		SF2	0.25	50	NM_SF2(0.25)_50	
				90	NM_SF2(0.25)_90	
	SSM	-	-	50	SSM_50	
		-	-	90	SSM_90	
		SF1	0.25	50	SSM_SF1(0.25)_50	
				90	SSM_SF1(0.25)_90	
Phase II	SSM	-	-	50	SSM_50	Container 2 and 3
		-	-	90	SSM_90	
		SF1	0.25	50	SSM_SF1(0.25)_50	
				90	SSM_SF1(0.25)_90	

**Table 7 materials-14-04005-t007:** Test result summary.

Test ID		Dynamic Yield Stress (Pa)			Viscosity (Pa·s)			Container
		w/o	w	Change Rate (%)	w/o	w	Change Rate (%)	
Phase 1	NM	15.73	15.15	−3.69	1.24	1.22	−1.61	1
	NM_SF1(0.25)_50	20.46	17.03	−16.76	1.35	1.29	−4.44	
	NM_SF1(0.25)_90	20.93	16.39	−21.68	1.30	1.28	−1.54	
	NM_SF1(0.50)_50	26.40	15.28	−42.14	2.13	1.28	−39.91	
	NM_SF1(0.50)_90	25.76	14.37	−44.22	2.15	1.36	−36.74	
	NM_SF2(0.25)_50	21.21	14.65	−30.93	1.85	1.26	−31.89	
	NM_SF2(0.25)_90	22.69	14.96	−34.07	1.91	1.27	−33.58	
	SSM_50	14.76	17.78	20.46	1.50	1.74	16.00	
	SSM_90	14.68	18.61	26.77	1.58	1.79	13.29	
	SSM_SF1(0.25)_50	18.98	20.13	6.06	1.71	1.80	5.26	
	SSM_SF1(0.25)_90	18.65	20.48	9.81	1.90	2.01	5.79	
Phase 2	SSM_50	14.85	16.78	13.00	1.58	1.72	8.86	2
	SSM_90	14.68	17.57	19.69	1.65	1.85	12.12	
	SSM_SF1(0.25)_50	19.05	20.68	8.56	1.95	1.96	0.51	
	SSM_SF1(0.25)_90	18.99	20.35	7.16	2.01	1.98	−1.49	
	SSM_50	15.01	14.41	−4.00	1.60	1.51	−5.63	3
	SSM_90	15.11	14.89	−1.46	1.71	1.52	−11.11	
	SSM_SF1(0.25)_50	18.79	14.14	−24.75	1.93	1.48	−23.32	
	SSM_SF1(0.25)_90	18.85	15.24	−19.15	1.99	1.53	−23.12	

## Data Availability

The data presented in this study are available on request from the corresponding author.

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
