# Peer review of "Changes in Rheological Properties of Mortars with Steel Slags and Steel Fibers by Magnetic Field"

_materials, 2021, doi:10.3390/ma14144005_

Round 1
Reviewer 1 Report
I recommend the paper to be published in its present form.
Reviewer 2 Report
Dear Authors,
thank you for your manuscript combining two interesting materials - mortar and steel - exposed to external magnetific field.
My comments are to the current verison are as follows:
- I propose to join some Figures to only one, for example instead of Figure 8 a-c to have only one Figure 8. Simillarly Figures 9-14. I think that it would keep clarity if you selecte suitable presentation
- I am not fully convinced about your conclusions supported by only two values (50 and 90V). From research viewpoint, more voltages would enhance the credit to your results.
- line 29: instead of [1,2,3] use [1-3]
Reviewer 3 Report
The problem of improvement of characteristics of steel fiber reinforced concrete is very important. The authors have proposed to use magnetic field for changing rheological properties of raw mix and alignment of steel fiber to get decrease of steel fiber seggregation. Through the magnetic field the adjusting of direction of steel fibers was achived and rheological properties of steel fiber raw mix were close to rheology of normal mix. Through the continuously change of the position of magnetic field reducing of viscosity of raw steel slag mix and steel slag mix with steeel fiber was achived. Results received can be applied to the tecjnology of steel fiber reinforced concrete.
There are suggestions for the authors:
- Mechanical characteristics of the hardened samples such as compressive and flexural strength should be given to show how the treatment of the raw mix with steel fiber by magnetic field resulted on its increasing.
- The energy consumption for magnetic field treatment per the volume of motar should be given.
Reviewer 4 Report
Introduction can be more succinct and to the point.
Line104: Needs to be reworded.
Line 181: What is the rationale behind using 50 V and 90 V for the experiment. The power supply can go up to 240v, so why not go higher or lower?
Table 6: The table is hard to follow. Additionally, In Phase II why were NM and SF2 excluded?
Figure 6 and Table 6 should appear after lines 227-232
Section 3.1 should be labeled as “Results of Phase I Test” for two reasons: 1) 3.2 is Results of Phase 2 Test, 2) section 3.1 not only discusses the results in NM but also discusses the results in SSM. The title is misleading.
Figure 12 c-d, both y-axes cannot be rate of change of yield stress
A deeper discussion of the work either in section 4 or section 5, is needed. Further interpretation of the data needs to be added along with the future plans or future for this technique.
Round 2
Reviewer 2 Report
Dear Authors,
thank you for taking my ideas into an account. If you have tried and it is not better then leave in in current form.
From my view point, the manuscript can be published in current form..